# Development and validation of an interpretable model integrating multimodal information for improving ovarian cancer diagnosis

Huiling Xiang ®[1,2,11], Yongjie Xiao[3,11], Fang Li[4,5,11], Chunyan Li[2], Lixian Liu[6], Tingting Deng[2], Cuiju Yan[2], Fengtao Zhou[7], Xi Wang[8,9], Jinjing Ou[2], Qingguang Lin[2], Ruixia Hong[4,5], Lishu Huang[4,5], Luyang Luo[7], Huangjing Lin[3,9], Xi Lin[2] ✉ & Hao Chen ®[7,10] ✉

Ovarian cancer, a group of heterogeneous diseases, presents with extensive characteristics with the highest mortality among gynecological malignances. Accurate and early diagnosis of ovarian cancer is of great significance. Here, we present OvcaFinder, an interpretable model constructed from ultrasound images-based deep learning (DL) predictions, Ovarian–Adnexal Reporting and Data System scores from radiologists, and routine clinical variables. Ovca-Finder outperforms the clinical model and the DL model with area under the curves (AUCs) of 0.978, and 0.947 in the internal and external test datasets, respectively. OvcaFinder assistance led to improved AUCs of radiologists and inter-reader agreement. The average AUCs were improved from 0.927 to 0.977 and from 0.904 to 0.941, and the false positive rates were decreased by 13.4% and 8.3% in the internal and external test datasets, respectively. This highlights the potential of OvcaFinder to improve the diagnostic accuracy, and consistency of radiologists in identifying ovarian cancer.

Ovarian cancer remains the most lethal gynecological cancer and accounted for approximately 14,070 cancer-related deaths and 22,240 new cases of cancer in the United States in 2018[1]. About 58% of ovarian cancers are initially diagnosed as metastatic ovarian cancers, which have a 5-year survival rate of only 30%, compared with a survival rate of 93% for localised cancers[2]. An accurate diagnostic method for the early diagnosis of ovarian cancer improves therapeutic outcomes by enabling early intervention. Patients with ovarian cancer who refer to gynaecology oncology centre for debulking surgery and systemic therapies have longer survival compared to those managed in

[1]Department of Radiology, State Key Laboratory of Oncology in South China, Guangdong Provincial Clinical Research Center for Cancer, Sun Yat-sen University Cancer Center, Guangzhou 510060, P. R. China. [2]Department of Ultrasound, State Key Laboratory of Oncology in South China, Guangdong Provincial Clinical Research Center for Cancer, Sun Yat-sen University Cancer Center, Guangzhou 510060, P. R. China. [3]AI Research Lab, Imsight Technology Co., Ltd., Nanshan, Shenzhen 518000, China. [4]Department of Ultrasound, Chongqing University Cancer Hospital, Chongqing, China. [5]Chongqing Key Laboratory for Intelligent Oncology in Breast Cancer (iCQBC), Chongqing University Cancer Hospital, Chongqing 400030, China. [6]Department of Ultrasound, Guangdong Second Provincial General Hospital, No. 466, Xingang Middle Road, Haizhu District, Guangzhou, Guangdong, China. [7]Department of Computer Science and Engineering, The Hong Kong University of Science and Technology, Hong Kong, China. [8]Zhejiang Lab, Hangzhou, China. [9]Department of Computer Science and Engineering, The Chinese University of Hong Kong, Hong Kong, China. [10]Department of Chemical and Biological Engineering, The Hong Kong University of Science and Technology, Hong Kong, China. [11]These authors contributed equally: Huiling Xiang, Yongjie Xiao, Fang Li. ✉e-mail: linxi@sysucc.org.cn; jhc@cse.ust.hk

community or general hospitals[3]. For patients with lesions of benign ultrasound morphology, the 2-year cumulative incidence of major complications, including invasive malignancy, torsion, and cyst rupture, was less than 0.5%, which can be followed up to prevent unnecessary surgeries as well as associated preoperative complications (~15%) and preserve fertility[4]. However, only 300,000 out of 2,000,000 women estimated to have exploratory surgery for a suspicious mass annually worldwide, were newly diagnosed with ovarian cancer[5,6], indicating the urgent need of a more accurate non-invasive diagnostic tool.

Compared with computed tomography (CT) and magnetic resonance imaging (MRI), transvaginal ultrasound (TVUS) is the most used diagnostic imaging tool for adnexal masses, for its lack of contraindications, low cost, and widespread availability. Various classification systems have been proposed but with limited acceptance, for the lack of standardised terminology or objective criteria. In contrast, the Ovarian–Adnexal Reporting and Data System (O-RADS) provides standardised terminology for lesion description and all risk categories with their corresponding management strategies, with the aim of improving diagnostic efficiency and realising tailored management[7].

Recently, deep-learning (DL) models have shown remarkable success in various diagnostic tasks. For example, DL has shown great promise in distinguishing papilloedema from other optic disc abnormalities in fundus photographs, with areas under the receiver operating characteristic curve (AUCs) ranging from 0.96 to 0.99[8], and in identifying breast cancer in mammography, where it outperformed five specialists with a mean increase in sensitivity of 14%[9]. For adnexal masses diagnosis, Zhang et al.[10] devised a ultrasound-based DL system, but it lacked additional external validation and clear clinicopathological information. Subsequently, Gao et al.[5] developed and validated another ovarian cancer diagnosis model using ultrasound images from 117,746 patients of 10 hospitals across China. They demonstrated the expert-level performance of their model and showed that it helped radiologists achieve significant improvements in diagnosis.

However, there remains room for improvement in the above-mentioned approaches. First, despite its high diagnostic performance in a wide range of diseases, DL is often criticized as a black box. In other words, it lacks transparency and explanation for its decisions, making it difficult for radiologists to understand what the DL models have learned from training images. Second, readily available clinical variables that may be of use in ovarian cancer diagnosis, such as the serum biomarker cancer antigen 125 (CA125), were not included in previously proposed DL models. The CA125 increases by 82% in patients with ovarian cancer and is widely used in clinical practice and screening programmes[11,12]. During the diagnostic process, multimodal information is generally needed before reaching the conclusion. However, to the best of our knowledge, there lacks studies that integrated multimodal information into an ovarian cancer risk stratification method.

Hence, the purpose of this study is to develop and validate the OvcaFinder to discriminate benign from ovarian cancer with the integration of ultrasound images-based DL predictions, assessments from radiologists, and routine clinical parameters. Our results show that OvcaFinder yields the highest performance when comparing with any single model or radiologists, with AUCs of 0.978 in the internal test dataset, and 0.947 in the external test dataset, respectively. OvcaFinder boosted the diagnostic performance of radiologists and decreased their false positive rates. In addition to identifying ovarian cancer, OvcaFinder is able to offer explanations to its predictions by highlighting the most important areas in heatmaps and reveal the impact of each parameter with Shapley values[13].

## Results
### Baseline information
As shown in Table 1, there were 3972 B-mode and colour Doppler ultrasound images of 296 (40.9%) benign and 428 (59.1%) malignant

**Table 1 | Demographic characteristics of the participants**

| | SYSUCC | | | CQUCC |
|---|---|---|---|---|
| | Training | Validation | Internal test dataset | External test dataset |
| No. of patients | 532 | 63 | 129 | 387 |
| Age (y) | | | | |
| Mean ± SD | 48 ± 12 | 46 ± 13 | 48 ± 14 | 43 ± 12 |
| Range | 16–82 | 16–69 | 20–79 | 18–83 |
| Menopausal status | | | | |
| Premenopausal | 291 (54.7) | 37 (58.7) | 70 (54.3) | 272 (70.3) |
| Postmenopausal | 241 (45.3) | 26 (41.3) | 59 (45.7) | 115 (29.7) |
| CA125 concentration | | | | |
| Mean ± SD | 1218 ± 3119 | 919 ± 1923 | 1274 ± 2902 | 245 ± 2413 |
| Range | 4–37,827 | 9–12,684 | 7–20,308 | 2–46,090 |
| Lesion diameter (mm) | | | | |
| Mean ± SD | 74.0 ± 36.5 | 74.8 ± 31.9 | 72.4 ± 32.9 | 71.2 ± 35.0 |
| Range | 10–224 | 30–161 | 15–164 | 19–334 |
| Histological type | | | | |
| Benign (%) | 215 (40.4) | 27 (42.9) | 54 (41.9) | 306 (79.1) |
| Teratoma | 72 (33.5) | 12 (44.5) | 20 (37.0) | 86 (28.1) |
| Endometriosis | 48 (22.3) | 9 (33.3) | 17 (31.4) | 102 (33.3) |
| Cyst | 33 (15.3) | 2 (7.4) | 5 (9.3) | 33 (10.8) |
| Cystadenoma | 19 (8.8) | 2 (7.4) | 4 (7.3) | 50 (16.4) |
| Inflammation | 17 (8.0) | 0 (0.0) | 2 (3.7) | 1 (0.3) |
| Corpus luteum | 11 (5.1) | 0 (0.0) | 1 (1.9) | 14 (4.7) |
| Thecoma | 6 (2.8) | 1 (3.7) | 1 (1.9) | 5 (1.6) |
| Fibroma | 4 (1.9) | 0 (0.0) | 1 (1.9) | 5 (1.6) |
| Hydrosalpinx | 2 (0.9) | 0 (0.0) | 2 (3.7) | 5 (1.6) |
| Other | 3 (1.4) | 1 (3.7) | 1 (1.9) | 5 (1.6) |
| Malignant (%) | 317 (59.6) | 36 (57.1) | 75 (58.1) | 81 (26.5) |
| Serous carcinoma | 237 (74.8) | 23 (63.9) | 60 (80.0) | 47 (58.0) |
| Borderline tumor | 30 (9.5) | 6 (16.7) | 5 (6.6) | 16 (19.7) |
| Endometrioid carcinoma | 9 (2.8) | 1 (2.8) | 2 (2.7) | 5 (6.2) |
| Clear cell | 9 (2.8) | 3 (8.3) | 2 (2.7) | 2 (2.5) |
| Other | 32 (10.1) | 3 (8.3) | 6 (8.0) | 11 (13.6) |

Data in parentheses are percentages. SD Standard deviation.

lesions from 724 patients in SYSUCC (mean age: 48 ± 13 years; range: 16–82 years). The lesion diameter ranged from 10 to 224 mm, with a mean diameter of 74.3 mm (standard deviation (SD): 35.5 mm). The concentration of CA125 ranged from 4 to 37,827 U/mL. These patients were randomly split into the training (2941 images of 532 lesions), validation (334 images of 63 lesions), and the internal test dataset (697 images of 129 lesions). In the external dataset, there were 2200 images from 387 patients (mean age: 43 ± 12 years; range: 18–83 years). The mean lesion diameter was 71.2 mm (SD: 35.0 mm). The concentration of CA125 ranged from 2 to 46,090 U/mL. Among 509 malignant lesions, there were 57 borderline tumors (11.2%). For malignant lesions. the average lesion diameter was 83.4 mm (range: 13–225 mm). Taking 35 U/mL as threshold, nearly 88.2% (449/509) patients had evaluated CA125 levels. Ascites and peritoneal thickening or nodules were found in 272 and 306 patients in ultrasound images, respectively.

### Performance of readers with O-RADS
After completing training, five readers showed high diagnostic performance in adnexal tumour classification. The O-RADS assessment scores were normalized into a range of 0 to 1, in order to calculate the performance of the AUCs. The average AUCs were 0.927 for the internal test dataset and 0.904 for the external dataset, respectively.

**Table 2 | Diagnostic performance of different models**

| Internal test dataset | Clinical model | Image based-DL predictions | OvcaFinder |
|---|---|---|---|
| AUC | 0.936 | 0.970 | 0.978 |
| | (0.902, 0.975) | (0.934, 0.993) | (0.953, 0.998) |
| p | 0.007 | 0.152 | Reference |
| Sensitivity (%) | 97.3 | 97.3 | 97.3 |
| | (93.3, 100.0) | (93.3, 100.0) | (93.3, 100.0) |
| p | 1.00 | 1.00 | Reference |
| Specificity (%) | 40.7 | 74.1 | 83.3 |
| | (28.3, 52.8) | (62.3, 84.9) | (73.6, 92.4) |
| p | $1.52 \times 10^{-5}$ | 0.062 | Reference |
| Accuracy (%) | 73.6 | 87.6 | 91.5 |
| | (68.0, 79.7) | (82.0, 92.2) | (86.7, 96.1) |
| PPV (%) | 69.5 | 83.9 | 89.0 |
| | (65.5, 74.8) | (78.5, 90.1) | (83.3, 94.9) |
| NPV (%) | 91.7 | 95.2 | 95.7 |
| | (79.2, 100.0) | (88.6, 100.0) | (88.9, 100.0) |
| External test dataset | | | |
| AUC | 0.842 | 0.893 | 0.947 |
| | (0.776, 0.895) | (0.855, 0.933) | (0.917, 0.970) |
| p | $4.65 \times 10^{-5}$ | $3.93 \times 10^{-6}$ | Reference |
| Sensitivity (%) | 85.2 | 88.9 | 88.9 |
| | (76.5, 92.6) | (81.5, 95.1) | (81.5, 95.1) |
| p | 0.581 | 1.000 | Reference |
| Specificity (%) | 53.3 | 68.6 | 90.5 |
| | (047.7, 58.5) | (64.0, 73.5) | (87.3, 93.8) |
| p | $2.21 \times 10^{-29}$ | $1.36 \times 10^{-20}$ | Reference |
| Accuracy (%) | 59.9 | 72.9 | 90.2 |
| | (55.3, 64.3) | (68.5, 77.3) | (87.1, 93.0) |
| PPV (%) | 32.5 | 42.9 | 0.713 |
| | (29.5, 35.7) | (38.6, 47.8) | (64.2, 79.1) |
| NPV (%) | 93.1 | 95.9 | 96.9 |
| | (89.8, 96.3) | (93.3, 98.2) | (94.8, 98.6) |

Data in parentheses are 95% confidence intervals; *DL* Deep learning, *AUC* Area under the receiver operating characteristic curve, *PPV* Positive predictive value, *NPV* Negative predictive value. We used an average value of O-RADS scores as the input factor of OvcaFinder. *p* values are for a comparison with OvcaFinder. The *p*-values of AUC were calculated using the function 'roc_test' in the python package of pROC. The *p*-values of sensitivity and specificity were calculated via two-sided McNemar test.

The readers showed a mean sensitivity of 96.2% and specificity of 73.3% in the internal dataset, and a mean sensitivity and specificity of 85.7% and 81.8%, respectively, in the external dataset.

**Performance of the image-based DL predictions**
DenseNet121, DenseNet169, DenseNet201, ResNet34, EfficientNet-b5, and EfficientNet-b6 achieved AUCs ranging from 0.898 to 0.923 in the internal test dataset and from 0.806 to 0.851 in the external test dataset (Supplementary Table 1) at the lesion level using B-mode and colour Doppler images, which was inferior to the final ensemble DL model. The ensemble model showed an AUC of 0.970, a sensitivity of 97.3%, and a specificity of 74.1% in the internal dataset. In the external dataset, the AUC was reduced to 0.893, the sensitivity was 88.9%, and the specificity was 68.6% (Table 2). As shown in Fig. 1, the red regions of the heatmaps contributed most to a given classification, while the blue regions were less important. To be more specific, areas with irregular solid components or projections on B-mode images were highlighted in the heatmap and were valuable features for malignancy prediction. With regard to colour Doppler images, the heatmap focused on areas with abundant angiogenesis. These were consistent with the

diagnostic criteria of ovarian tumors in clinical practice. For benign lesions, there were 27.8% (15/54) and 19.8% (60/306) cases with hotspots shown in the internal and external test datasets, respectively. As for cancerous lesions, a percentage of 4.0% (3/75) in the internal test dataset and 12.3% (10/81) in the external test dataset were observed without hotspots displayed, respectively.

**Performance of the clinical model**
In the internal test dataset, the clinical model achieved an AUC of 0.936, a sensitivity of 97.3%, and a specificity of 40.7%. Within the external test dataset, the clinical model yielded an AUC of 0.842, a sensitivity of 85.2%, and a specificity of 53.3% (Table 2).

**Performance of the OvcaFinder**
As shown in Fig. 2, with the integration of clinical information, O-RADS scores, and image-base DL predictions, OvcaFinder showed higher performance (AUC: 0.978 [95% CI: 0.953, 0.998]) than the clinical model (AUC: 0.936, $p = 0.007$) and image-based DL predictions (AUC: 0.970, $p = 0.152$) in the internal test dataset. OvcaFinder, with an AUC of 0.947 (95% CI: 0.917, 0.970) also outperformed the clinical model (AUC: 0.842, $p = 4.65 \times 10^{-5}$) and image-based DL predictions (AUC: 0.893, $p = 3.93 \times 10^{-6}$) on the external test dataset. For a fair comparison, we compared the specificities of three models via keeping similar sensitivities. With the internal test dataset, when the sensitivity was maintained at 97.3%, OvcaFinder showed a higher specificity (83.3%) than the clinical model (40.7%, $p = 1.52 \times 10^{-5}$) and DL predictions (74.1%, $p = 0.062$). On the external test dataset, while maintaining a similar sensitivity to other models, OvcaFinder showed a specificity of 90.5%, which outperformed the clinical model (53.3%, $p = 2.21 \times 10^{-29}$) and image-based DL predictions (68.6%, $p = 1.36 \times 10^{-20}$; Table 2). We observed that (Fig. 3) the image-based DL predictions weighed the most importantly regarding the decision prediction of OvcaFinder, followed by O-RADS, CA125 concentration, patient's age, and lesion diameter.

In the reader study, the AUCs of readers ranged from 0.900 to 0.958. But with the aid of OvcaFinder, the AUCs were substantially increased, ranging from 0.971 to 0.981 with the internal test dataset, without any decrease on sensitivity. Similar improvements were observed for all readers on the external test dataset. Moreover, OvcaFinder boosted the readers' diagnostic accuracy with fewer false positives (Fig. 4, and Table 3). The average false positive rate decreased from 26.7% (range: 13.0–38.9%) to 13.3% (range: 7.4–18.5%, p = 0.029) and from 18.2% (range: 10.8–29.4%) to 9.9% (range: 8.2–12.4%, $p = 0.033$) on the internal and external test datasets, respectively, which would potentially obviate the need for unnecessary biopsies or surgeries.

The inter-reader agreement for ovarian cancer diagnosis were summarized in Table 4. Inter-reader kappa values ranged from 0.711 to 0.924 and from 0.588 to 0.796 in the internal and external test dataset, respectively, indicating fair to excellent agreement. With OvcaFinder, the inter-reader kappa values improved to 0.886 to 0.983 in the internal test dataset and 0.863 to 0.933 on the external test dataset, suggesting excellent agreement.

## Discussion
Ovarian cancer is a group of heterogeneous disease with highly complex features. Differential diagnosis before surgery requires the integration of multimodal information. The diagnostic values of image-based DL predictions, O-RADS scores from readers, and clinical parameters in ovarian cancer diagnosis, have been explored previously. However, little is known about the capacity of combining multimodal features to improve diagnosis. Here, we developed OvcaFinder by integrating image-based DL predictions, readers' assessments, and clinical parameters, for the identification of ovarian cancer. OvcaFinder outperformed the image-based DL model, clinical model, and

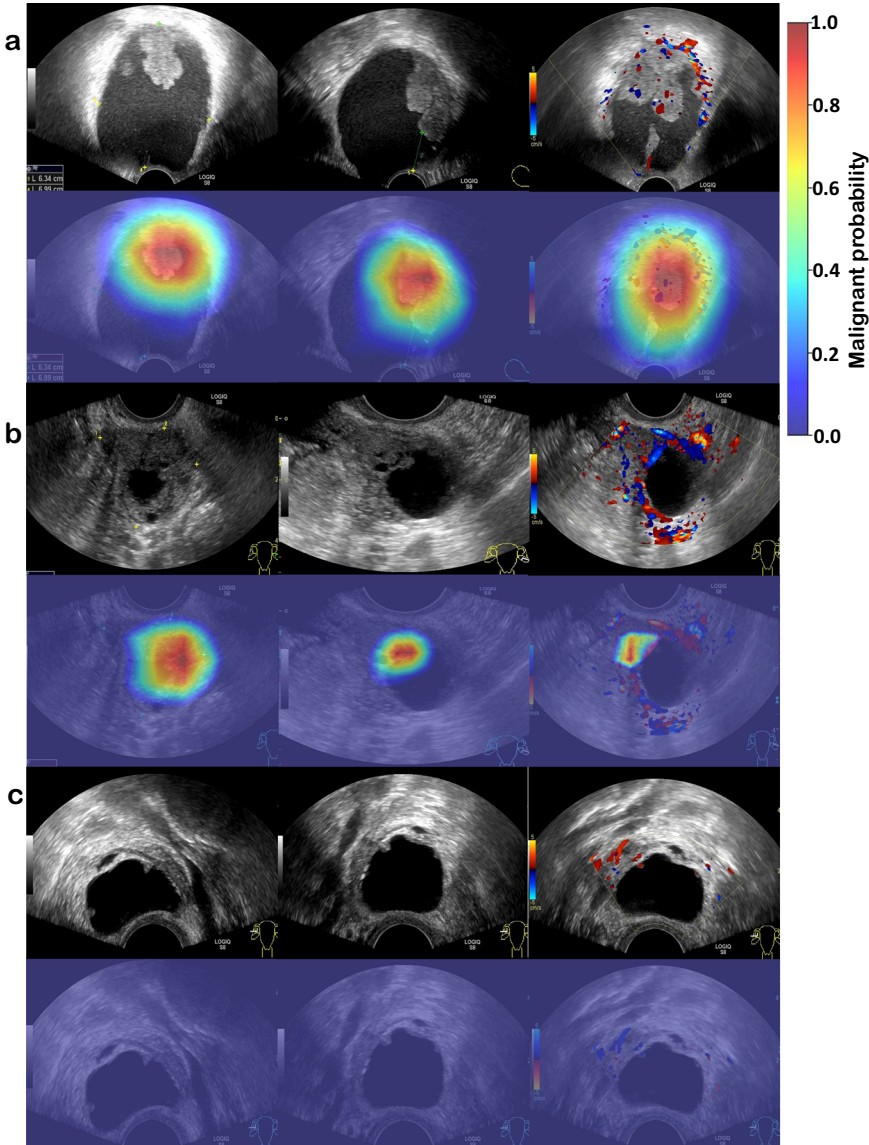

**Fig. 1 | Heatmap visualisation of image-based deep learning predictions of malignancy.** Visual explanations of DL models are definitely important for qualitative review and clinical relevance, namely irregular solid components, projections, and areas with abundant blood flow signals. **a** Carcinosarcoma of a 44-year old female; **b** high-grade serous carcinoma of a 65-year old female; and (**c**) hydrosalpinx of a 49-year old female that was misdiagnosed by all readers but showed a low probability of malignancy in the heatmap. In the first row of each case, the first two images are B-mode images, and the following one is the colour Doppler image. The images in the second row are their corresponding heatmaps.

readers, achieving AUCs of 0.978 and 0.947 in the internal and external test datasets, respectively. Without reducing sensitivities, OvcaFinder significantly improved the performances of readers with an increase of 5% and 3.8% in mean AUCs and a reduction of 13.4% and 8.3% in the false positive rate in the internal and external test datasets, respectively. Improvements in inter-reader agreement were also observed. These results highlight the potential of OvcaFinder to serve as a non-invasive tool to improve the accuracy, and consistency of radiologists in distinguishing benign from malignant ovarian lesions and reducing the number of false positives.

A strength of this study is that we used the O-RADS scoring system in the reader study to ensure accurate and reproducible assessments. The external validations in previous studies have suggested that O-RADS performed well, with AUCs ranging from 0.90 to 0.98[14–17], thereby confirming the feasibility of using O-RADS in our model. In our study, the readers achieved high-level performance, with AUCs of 0.927 and 0.904 in the internal and external test datasets, respectively. The sensitivities of readers in the internal test dataset were inferior to those in the external test dataset. This difference may be explained by distribution shift due to factors like relatively higher proportion of typical cases with heavier tumor burden in the internal test dataset, as evidenced by significantly higher CA125 levels ($p < 0.0001$)[18].

Here, we develop and evaluate a multimodal ovarian cancer analysis model (OvcaFinder) that comprises routinely available clinical information, radiologists' assessments, and DL predictions. OvcaFinder achieved high performance in both the internal and external test datasets. As shown in Fig. 3, we found that CA125, together with lesion diameter and the patient's age, did provide additional benefits in tumour diagnosis. We also developed an image-based DL model that achieved an AUC of 0.970 with the internal dataset but only 0.893 with the external dataset, which showed that the generalisability of the image-based DL model in real-world setting was confined[19,20]. OvcaFinder correctly identified cases that it has never seen before (AUC: 0.893 vs. 0.947, $p = 3.93 \times 10^{-6}$), suggesting higher generalisability on external data. Chen et al.[21] constructed a DL model and showed that their model had comparable diagnostic accuracy to expert subjective

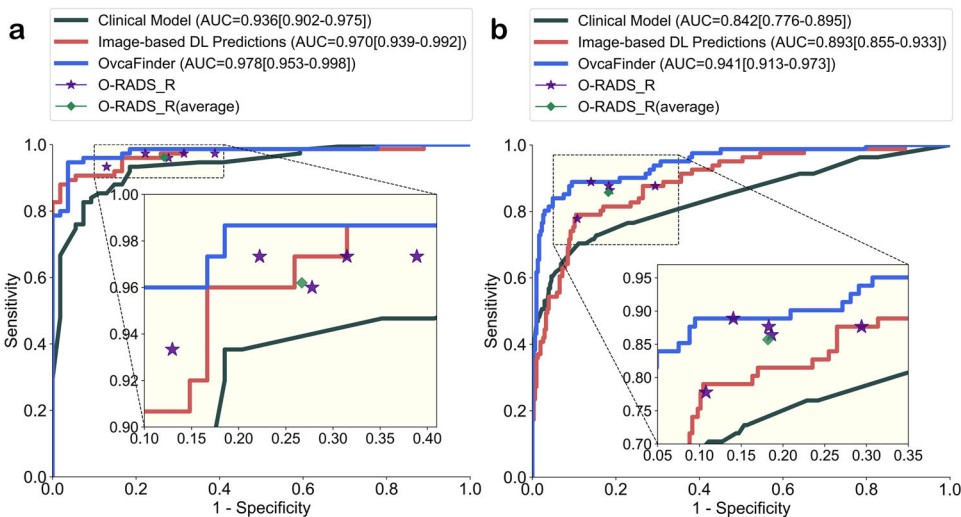

**Fig. 2 | Receiver operator characteristic curves of the three models and the readers. a** Internal test dataset; and **b** external test dataset. O-RADS Ovarian-adnexal reporting and data system, *R* Radiologist.

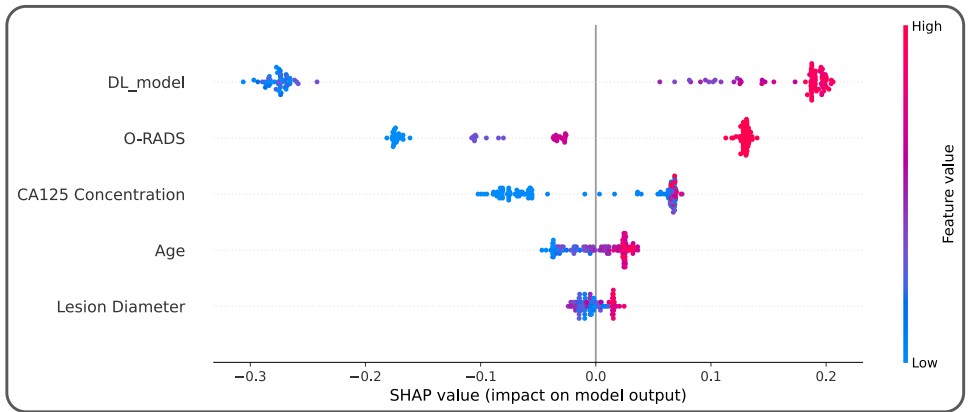

**Fig. 3 | Global Shapley values for the interpretation of OvcaFinder.** The horizontal ordinate represents the mean absolute Shapley value, indicating the global importance weights of the features. Each dot represents an individual patient (*n* = 129). The higher the Shapley values, the greater probability of ovarian cancer.

assessments and O-RADS assessments in a single-centre setting. Gao et al. (9) showed that DL improved the performance of radiologists. However, these studies did not clarify how DL could be used to streamline workflows. In this study, we showed that OvcaFinder using multimodal information significantly outperformed all readers (*p* < 0.05) with improved inter-reader agreement. Specifically, it reduced the false positive rates by 13.4% and 8.3% in internal and external datasets, respectively, while maintaining similar sensitivities.

Efforts were also made to enhance the interpretability of Ovca-Finder. Most DL models built previously for adnexal tumour diagnosis from ultrasound images did not show the most important features or areas that were highly relevant to their final classification, which hinders the building of trust that readers have in DL models[5,10]. Here, we found that heatmaps facilitated the assessment of adnexal masses by highlighting areas with irregular solid components, projections, or abundant blood signals, which is in accordance with current guidelines (Fig. 1)[7,22,23]. However, we observed some benign lesions (27.8% in the internal test dataset and 19.8% in the external test dataset) were also highlighted in the heatmaps. These lesions often contained thick septations or were adjacent to normal ovarian tissues, which needs to be further optimized by enrolling more healthy controls and benign cases. In addition, local and global Shapley values demonstrated the relative contributions of each modality in OvcaFinder on individual

patient and cohort, respectively. We observed that the features of image-based DL prediction clearly had the greatest overall effect on the decision made by OvcaFinder. Moreover, O-RADS scores also made a large contribution. CA125 concentration, the patient's age, and the lesion diameter had less of an influence on the decision. Abnormal CA125 levels could be found in 5% of patients with menstruation or benign diseases, such as endometriosis, which might partially explain why CA125 showed less contribution than OvcaFinder[24]. Timmerman et al.[25] also reported that CA125 was less informative than ultrasound in ovarian cancer diagnosis.

We acknowledge the limitations of this study. First, there might be a selection bias in this retrospective study. Pathology-proven adnexal tumors from two cancer hospitals were enrolled, which resulted in a relatively higher malignancy rate than usual. The applicability of the strategy to lower risk populations where the prevalence of cancer is low remains to be determined. A large-scale dataset, containing pathology-proven lesions, healthy controls and followed-up cases, not only from cancer hospitals but also general hospitals, would be useful for validating the OvcaFinder in a prospective setting to confirm its reliability. Second, other factors including personal history, family history, BRCA mutations, and the use of hormone replacement therapy were also of importance in the risk classification[1,26]. In the future study, we will further explore the added value of such factors. Third, other

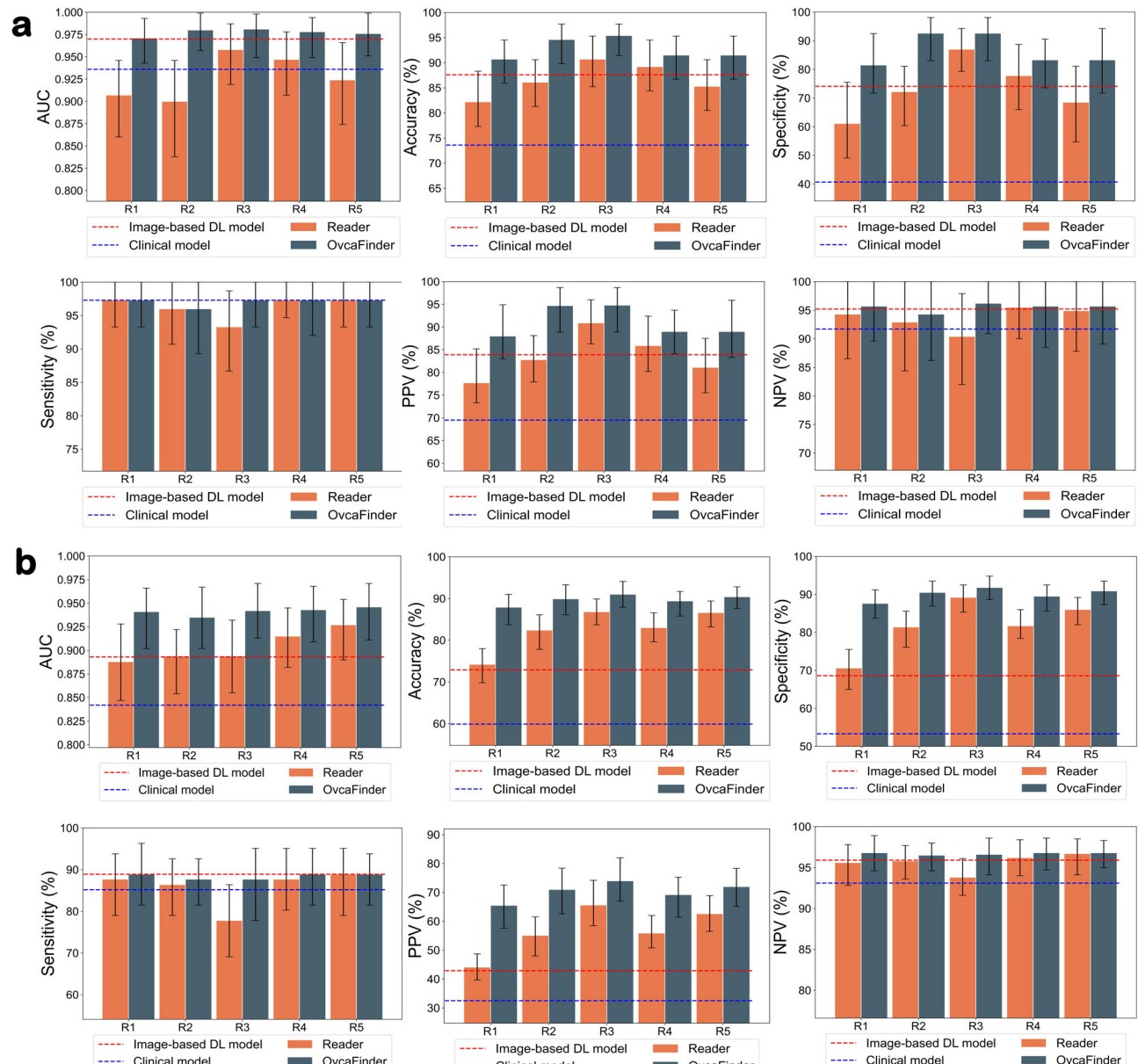

**Fig. 4 | Performance of image-based deep learning model, clinical model, readers, and the OvcaFinder.** We reported the observed values (measure of centre) and 95% confidence intervals (error bars) of the AUCs, accuracies, specificities, sensitivities, PPVs, and NPVs of five readers (R1-R5), the image-based deep learning model, the clinical model, and OvcaFinder in the internal (**a**, n = 129) and external (**b**, n = 387) test datasets. OvcaFinder improved the readers' AUCs, accuracies, specificities, and PPVs, while maintaining equivalent sensitivities and NPVs. AUC Area under the receiver operating characteristic curve, PPV Positive predictive value, NPV Negative predictive value.

imaging examinations such as CT, MRI, PET-CT, also play important roles in ovarian cancer diagnosis, and combining the information from these modalities could potentially further improve the performance of OvcaFinder.

In this study, we present clear evidence for the utility of the interpretable OvcaFinder in ovarian cancer diagnosis. OvcaFinder integrated ultrasound images, clinical information, and interpretations from radiologists and achieved the highest performance in both internal and external datasets, which highlighted the necessity of multimodal information integration for automatic ovarian cancer diagnosis. By analysing heatmaps and Shapley values, the decisions of OvcaFinder can be further explained, and the importance of each feature can be revealed. Using OvcaFinder led to significant improvements in radiologists' performance, increase in inter-reader agreement, and

reduction in the false positive rate, indicating potential for real-world usage as a promising non-invasive assistant tool.

## Methods

### Study design and participants

The study protocol was approved by Sun Yat-sen University Cancer Center's Institutional Review Board (B2022-112-01), with a waiver of the requirement for informed consent due to its retrospective nature. Patients were eligible if they presented with at least one pathology-proven adnexal lesion visible on TVUS examination. To ensure a complete evaluation, transabdominal examinations were included if the lesions were too large to be fully evaluated by TVUS. When multiple lesions were detected, the lesion with the most complex morphology was chosen for analysis. If lesions had similar features, the largest

**Table 3 | Diagnostic performance of OvcaFinder and human readers using O-RADS**

| | Reader A | | Reader B | | Reader C | | Reader D | | Reader E | |
|---|---|---|---|---|---|---|---|---|---|---|
| **Internal dataset** | O-RADS | OvcaFinder | O-RADS | OvcaFinder | O-RADS | OvcaFinder | O-RADS | OvcaFinder | O-RADS | OvcaFinder |
| AUC | 0.907 (0.860, 0.946) | 0.971 (0.943, 0.993) | 0.900 (0.838, 0.946) | 0.980 (0.957, 0.999) | 0.958 (0.919, 0.987) | 0.981 (0.949, 0.998) | 0.947 (0.907, 0.978) | 0.978 (0.949, 0.994) | 0.924 (0.874, 0.966) | 0.976 (0.951, 0.999) |
| $p$ | 0.002 | | $1.50 \times 10^{-3}$ | | 0.120 | | 0.056 | | 0.007 | |
| Sensitivity (%) | 97.3 (93.3, 100.0) | 97.3 (93.3, 100.0) | 96.0 (90.7, 100.0) | 96.0 (89.3, 100.0) | 93.3 (86.7, 98.7) | 97.3 (93.3, 100.0) | 97.3 (94.7, 100.0) | 97.3 (92.0, 100.0) | 97.3 (93.3, 100.0) | 97.3 (93.3, 100.0) |
| $p$ | 1.00 | | 1.00 | | 0.375 | | 1.00 | | 1.00 | |
| Specificity (%) | 61.1 (49.1, 75.5) | 81.5 (71.7, 92.5) | 72.2 (60.4, 81.1) | 92.6 (83.0, 98.1) | 87.0 (79.3, 94.3) | 92.6 (83.0, 98.1) | 77.8 (66.0, 88.7) | 83.3 (73.6, 90.6) | 68.5 (54.7, 81.1) | 83.3 (71.7, 94.3) |
| $p$ | 0.013 | | $9.77 \times 10^{-4}$ | | 0.375 | | 0.549 | | 0.057 | |
| Accuracy (%) | 82.2 (77.3, 88.3) | 90.7 (85.9, 94.5) | 86.1 (81.3, 90.6) | 94.6 (89.8, 97.7) | 90.7 (85.2, 95.3) | 95.4 (91.4, 97.7) | 89.2 (84.4, 94.5) | 91.5 (86.7, 95.3) | 85.3 (80.5, 90.6) | 91.5 (86.7, 95.3) |
| PPV (%) | 77.7 (73.3, 85.2) | 88.0 (83.0, 94.9) | 82.8 (77.9, 88.1) | 94.7 (88.9, 98.7) | 90.9 (86.3, 96.0) | 94.8 (88.9, 98.7) | 85.9 (80.2, 92.4) | 89.0 (84.1, 93.7) | 81.1 (75.5, 87.5) | 89.0 (83.3, 95.9) |
| NPV (%) | 94.3 (86.5, 100.0) | 95.7 (89.6, 100.0) | 92.9 (84.4, 100.0) | 94.3 (86.2, 100.0) | 90.4 (82.0, 97.9) | 96.2 (90.9, 100.0) | 95.5 (90.0, 100.0) | 95.7 (88.5, 100.0) | 94.9 (87.8, 100.0) | 95.7 (89.1, 100.0) |
| **External test dataset** | | | | | | | | | | |
| AUC | 0.888 (0.847, 0.928) | 0.941 (0.902, 0.966) | 0.894 (0.854, 0.922) | 0.935 (0.902, 0.967) | 0.894 (0.855, 0.932) | 0.942 (0.913, 0.971) | 0.915 (0.882, 0.945) | 0.943 (0.909, 0.968) | 0.927 (0.890, 0.954) | 0.946 (0.911, 0.971) |
| $p$ | 0.006 | | 0.005 | | 0.008 | | 0.025 | | 0.149 | |
| Sensitivity (%) | 87.7 (79.0, 93.8) | 88.9 (81.5, 96.3) | 86.4 (79.0, 92.6) | 87.7 (81.5, 92.6) | 77.8 (69.1, 86.4) | 87.7 (77.8, 95.1) | 87.7 (80.3, 95.1) | 88.9 (81.5, 95.1) | 88.9 (79.0, 95.1) | 88.9 (81.5, 93.8) |
| $p$ | 1.00 | | 1.00 | | 0.077 | | 1.00 | | 1.00 | |
| Specificity (%) | 70.6 (65.0, 75.5) | 87.6 (83.7, 91.2) | 81.4 (76.1, 85.6) | 90.5 (86.9, 93.5) | 89.2 (85.3, 92.5) | 91.8 (88.6, 94.8) | 81.7 (78.4, 86.0) | 89.5 (85.6, 92.5) | 86.0 (82.0, 89.2) | 90.9 (87.3, 93.5) |
| $p$ | $3.07 \times 10^{-12}$ | | $8.36 \times 10^{-6}$ | | 0.185 | | $6.96 \times 10^{-5}$ | | 0.004 | |
| Accuracy (%) | 74.2 (69.8, 78.0) | 87.9 (83.7, 91.0) | 82.4 (77.8, 86.1) | 89.9 (86.1, 93.3) | 86.8 (83.7, 89.9) | 91.0 (87.9, 94.1) | 83.0 (79.6, 86.6) | 89.4 (85.8, 91.7) | 86.6 (83.2, 89.4) | 90.4 (87.6, 92.8) |
| PPV (%) | 44.1 (39.7, 48.7) | 65.5 (57.6, 72.6) | 55.1 (48.0, 61.5) | 71.0 (62.6, 78.4) | 65.6 (58.5, 74.2) | 74.0 (67.0, 82.0) | 55.9 (50.8, 62.0) | 69.2 (61.4, 75.3) | 62.6 (56.5, 68.9) | 72.0 (65.2, 78.3) |
| NPV (%) | 95.6 (92.8, 97.8) | 96.8 (94.6, 98.9) | 95.8 (93.6, 97.7) | 96.5 (94.6, 98.0) | 93.8 (91.6, 96.1) | 96.6 (94.1, 98.6) | 96.2 (94.0, 98.4) | 96.8 (94.7, 98.6) | 96.7 (94.1, 98.5) | 96.8 (95.0, 98.3) |

Data in parentheses are 95% confidence intervals; O-RADS Ovarian-Adnexal Reporting and Data System, AUC Area under the receiver operating characteristic curve, PPV Positive predictive value, NPV Negative predictive value. The p-values of AUC were calculated using the function 'roc_test' in the python package of pROC. The p-values of sensitivity and specificity were calculated via two-sided McNemar test.

**Table 4 | Inter-reader Agreement (Kappa Coefficients) for Diagnostic Performance**

|  |  | Readers | Internal test dataset | External test dataset |
|---|---|---|---|---|
| Inter-reader | O-RADS | A vs. B | 0.760 (0.640, 0.873) | 0.649 (0.576, 0.720) |
|  |  | A vs. C | 0.711 (0.601, 0.817) | 0.588 (0.521, 0.653) |
|  |  | A vs. D | 0.837 (0.739, 0.929) | 0.693 (0.627, 0.763) |
|  |  | A vs. E | 0.924 (0.844, 0.982) | 0.667 (0.602, 0.736) |
|  |  | B vs. C | 0.800 (0.679, 0.901) | 0.669 (0.597, 0.742) |
|  |  | B vs. D | 0.826 (0.711, 0.914) | 0.742 (0.673, 0.812) |
|  |  | B vs. E | 0.802 (0.694, 0.909) | 0.760 (0.685, 0.828) |
|  |  | C vs. D | 0.835 (0.741, 0.932) | 0.681 (0.603, 0.760) |
|  |  | C vs. E | 0.782 (0.679, 0.878) | 0.760 (0.687, 0.826) |
|  |  | D vs. E | 0.911 (0.829, 0.982) | 0.796 (0.732, 0.856) |
|  | OvcaFinder | A vs. B | 0.886 (0.798, 0.951) | 0.869 (0.812, 0.921) |
|  |  | A vs. C | 0.902 (0.834, 0.968) | 0.868 (0.815, 0.925) |
|  |  | A vs. D | 0.983 (0.949, 1.000) | 0.910 (0.862, 0.954) |
|  |  | A vs. E | 0.983 (0.949, 1.000) | 0.896 (0.841, 0.940) |
|  |  | B vs. C | 0.952 (0.889, 1.000) | 0.863 (0.804, 0.917) |
|  |  | B vs. D | 0.902 (0.840, 0.967) | 0.894 (0.839, 0.941) |
|  |  | B vs. E | 0.902 (0.825, 0.967) | 0.906 (0.854, 0.952) |
|  |  | C vs. D | 0.918 (0.855, 0.983) | 0.892 (0.840, 0.940) |
|  |  | C vs. E | 0.918 (0.855, 0.983) | 0.891 (0.834, 0.939) |
|  |  | D vs. E | 0.967 (0.916, 1.000) | 0.933 (0.893, 0.973) |

Data in parentheses are 95% confidence intervals; *O-RADS* Ovarian-Adnexal Reporting and Data System.

one was included. Anonymised clinicopathologic information and ultrasound findings were obtained from the password-protected database. Women aged more than 50 years were defined as post-menopausal. The exclusion criteria were: (1) physiological changes, such as a follicle or corpus luteum with a diameter less than 3 cm in premenopausal women; (2) a prior diagnosis of ovarian cancer; (3) loss of clinicopathologic information; or (4) a time interval between ultrasound examination and biopsy or surgery exceeding 120 days. Borderline tumours were assigned to the malignancy group[5,21,27].

### Image collection and reader study
B-mode and colour Doppler images were acquired by using commercially available equipment, including GE Logiq S8 (GE Healthcare, Milwaukee, WI, USA) or Aplio 300, 400, or 500 (Toshiba Medical System, Tokyo, Japan) systems. We retrospectively collected 3972 images of 724 lesions from patients in Sun Yat-sen University Cancer Center (SYSUCC) from February 2011 to May 2021. We randomly divided these images into training, validation, and internal test datasets at a ratio of 7:1:2. The external validation dataset was composed of 2200 images of 387 lesions obtained from patients in Chongqing University Cancer Hospital from December 2018 to June 2021.

Readers A, B, C, D, and E had 2, 3, 5, 9, and 19 years of experience, respectively. Blinded to any clinicopathologic information, they were trained in feature description and lesion categorisation using 60 additional cases. The trained readers then independently assessed all anonymised and randomised lesions, and assigned each lesion one of the following O-RADS risk scores[7]: 2, almost certainly benign (<1% risk of malignancy); 3, low risk of malignancy (1–10%); 4, intermediate risk of malignancy (10–50%); and 5, high risk of malignancy (≥50%).

### Model construction
**Image-based DL model.** The image-based DL model was designed to identify ovarian cancer based only on ultrasound images, without any additional information. The proposed image-based DL model was an ensembled model of six different backbones: DenseNet121[28], DenseNet169[28], DenseNet201[28], ResNet34[29], EfficientNet-b5[30], and

EfficientNet-b6[30]. Specifically, all models were first initialised with ImageNet[31] pretrained weights and then fine-tuned on our training dataset. All models have the same training configurations as follows. We used Adam as the optimizer with a learning rate of 0.0001. The input image resolution was set to 512 × 512, and the batch sizes was set to 8. The models were trained 100 epochs on the training dataset and validated after every epoch on the validation dataset. During training, several data augmentation strategies were used to increase the generalization ability of the model, including random horizontal flipping, rotation, and colour jitter operation. We selected the weights with the best performance of AUC on the validation dataset as the final weights for each model. Finally, we ensembled the predictions of the six models by averaging their predicted probabilities as the final score. The code was developed using the public framework PyTorch on a workstation equipped with two NVIDIA TITAN Xp graphic processing units.

### OvcaFinder and clinical model
The OvcaFinder and the clinical model were constructed based on Random Forest (RF) algorithm. OvcaFinder was a multimodal information-based model with human in the loop. Three clinical factors (patient's age, lesion diameter, and CA125 concentration), O-RADS scores diagnosed by readers, and DL-based predictions were used to build the input with 5-dim vectors to develop OvcaFinder (Fig. 5). Moreover, the clinical model only used three aforementioned clinical factors to build the input with 3-dim vectors during the model development. Specifically, The RF models were set to train with N estimators. For each estimator, we use Bootstrapping method to randomly resample the training set with replacement 1000 times to create simulated datasets. A simulated dataset was used to grow a decision tree. Therefore, we obtained a forest of N decision trees with different structures, as the trees were developed using different simulated datasets. The majority voting algorithm was then used to combine the predictions of each decision tree to generate the final output. For the OvcaFinder and the clinical model, we both developed 291 RF models with different numbers of estimators ranging from 10 to 300. Finally,

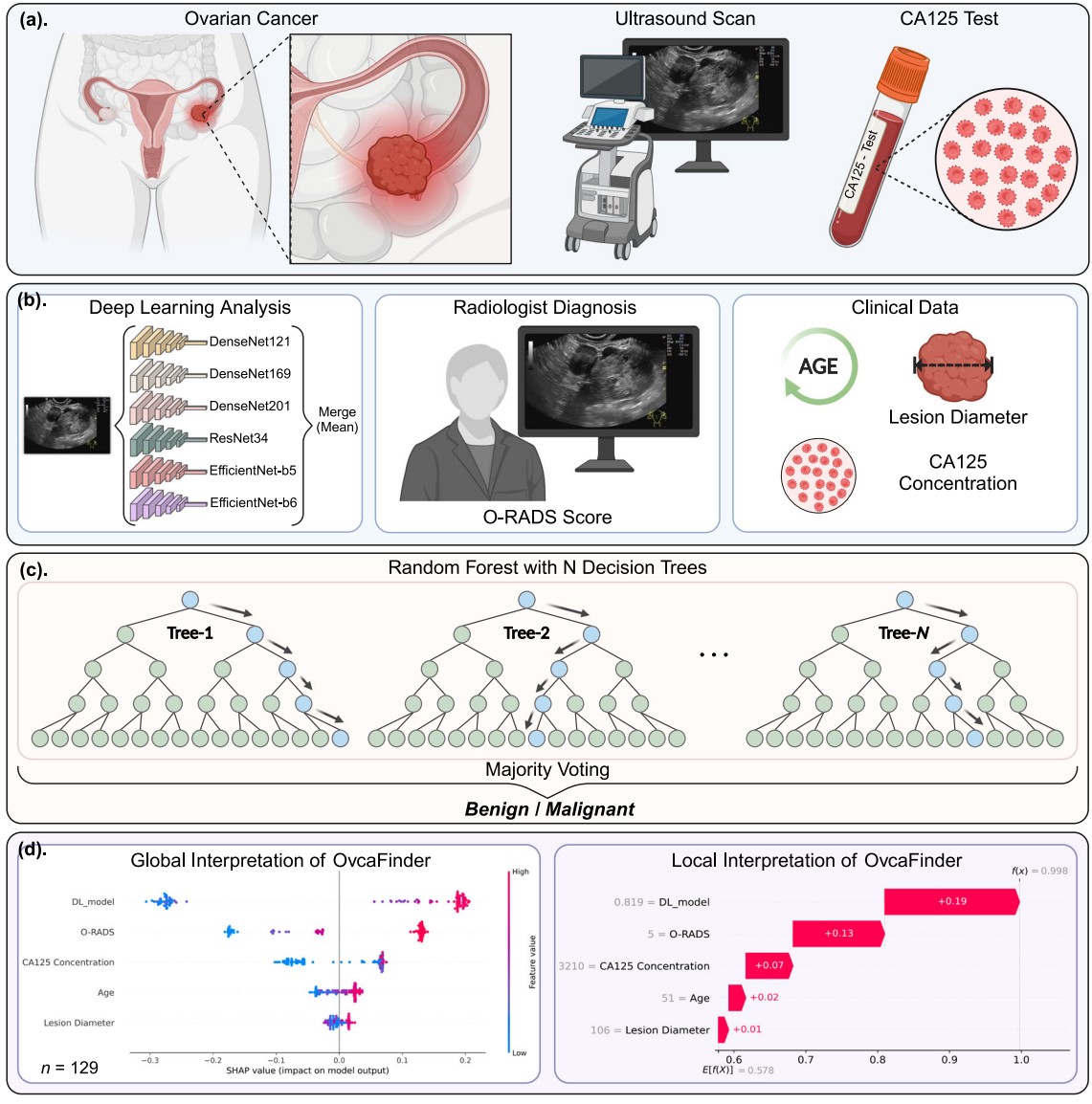

**Fig. 5 | Development of OvcaFinder using image-based deep learning predictions, O-RADS scores from radiologists, and clinical parameters. a** Data acquisition; **b** multimodal information; **c** model development; and **d** global and local interpretation of OvcaFinder (*n* = 129). O-RADS Ovarian-adnexal reporting and data system.

we found that N = 70 for the OvcaFinder and N = 20 for the clinical model would lead the models achieve the best performance of AUC on the validation dataset.

**Interpretation of OvcaFinder**

Heatmaps and Shapley values were used to enhance the interpretability of OvcaFinder at both the image and feature levels. To allow a clear visual understanding of the underlying basis of image-based malignancy prediction, we used the gradient-weighted class activation mapping[32] technique. Specifically, after feeding an image into the well-trained CNN model, we extracted feature maps with multiple channels of the final convolutional layer through the forward propagation. Also, we obtained the gradient weights, that contained the importance of each channel, by using the final prediction score of ovarian cancer to calculate the gradient information back to the final convolutional layer through back propagation. Then, we multiplied feature maps and gradient weights to generate the weighted combination of feature maps. Finally, we generated the heatmap by averaging the feature maps into one channel and

resizing it to the original resolution of the input image. We then averaged six heatmaps into one, as the model was an ensemble model of six backbones.

Furthermore, the Shapley values were used to calculate the specific contribution rank on each input feature of OvcaFinder. Local Shapley values were calculated for individual features of each instance (Fig. 6) to demonstrate the interpretability of OvcaFinder in terms of how the model decided for an individual sample. First, the expected value (mean value) was estimated for OvcaFinder's decision probabilities for all training samples and was set as the base value. The local Shapley values of all given features were then added to the base value to construct the final decision probability. Global Shapley values, which indicated the average impact of each feature on the magnitude of the model output, were computed by averaging the absolute local Shapley values across all instances.

**Statistical analysis**

Diagnostic performance was evaluated by calculating the AUC, accuracy, sensitivity, specificity, positive predictive value, and negative

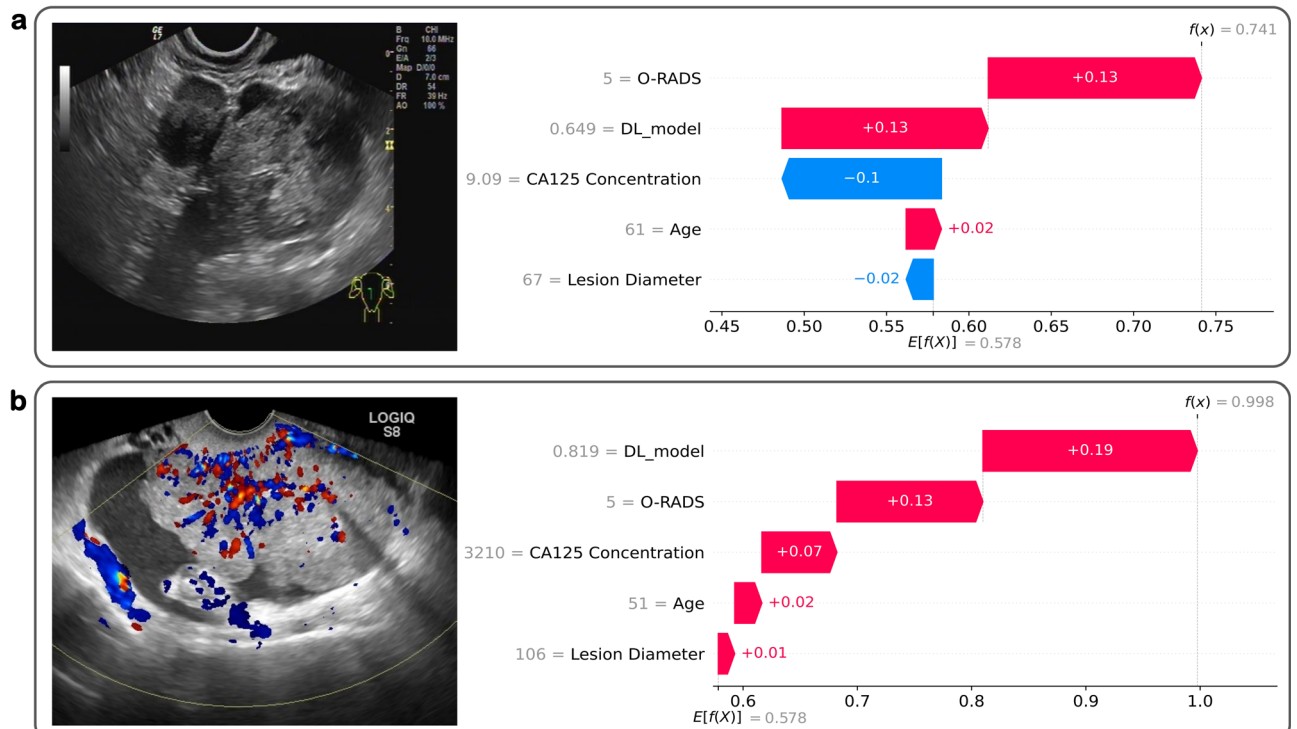

**Fig. 6 | Local Shapley values for the interpretation of OvcaFinder.** The horizontal ordinate represents the decision score. $f$ is the function of OvcaFinder, $f(x)$ is the final decision probability for input $x$, and $E[f(X)]$ is the expected value (mean value) for OvcaFinder's decision probabilities of all training samples (which is 0.578). For sample (**a**), the age, DL model prediction, and O-RADS score values were 61 years, 0.649 (prediction probability) and 5, giving +0.02, +0.13, and +0.13 contributions supporting the decision to apply a malignant label, respectively. In addition, the lesion diameter and CA 125 concentration values were 67 mm and 9.09 U/mL, giving −0.02 and −0.1 negative impacts against the decision to apply a malignant label, respectively. These contributions were then added to the expected value to obtain the final decision probability of 0.741. For sample (**b**), the lesion diameter, age, CA125 concentration, O-RADS score, and DL model prediction values were 106 mm, 51 years, 3210 U/mL, 5 and 0.819 (prediction probability), giving +0.01, +0.02, +0.07, +0.13, and +0.19 contributions supporting the decision to apply a malignant label. These contributions were then added to the expected value to obtain the final decision probability of 0.998. DL deep learning, O-RADS Ovarian-adnexal reporting and data system.

predictive value with 95% confidence intervals (CIs). The 95% CIs were calculated using the nonparametric bootstrap method with 1000 resampling events, while keeping a constant ratio of positive and negative cases. The mean AUC of five readers was calculated by averaging their AUC values. Comparisons were made between the performance of the models and readers in both the internal and external test datasets. We calculated $p$ values to determine significant differences between different models, or between the OvcaFinder and the readers, using the pROC library in R (version 3.6.3) for AUCs and McNemar's test for sensitivities and specificities. Interobserver agreement were assessed using Cohen kappa values, which were interpreted as follows: 0.21–0.40, fair; 0.41–0.60, moderate; 0.61–0.80, substantial; 0.81–1.00, excellent[33]. Two-tailed $p < 0.05$ were considered statistically significant.

**Reporting summary**

Further information on research design is available in the Nature Portfolio Reporting Summary linked to this article.

## Data availability

The original ultrasound images and clinical data used in this study are not publicly available due to the restrictions of hospital regulations and patient privacy. All data supporting the findings of this study are available on requests for non-commercial purposes from the corresponding authors X.L. and H.C. typically within two weeks. The data generated in this study for creation of figures and tables are provided with this paper. Source data are provided with this paper.

## Code availability

The codes of the proposed model in this study have been deposited at github (https://github.com/Xiao-OMG/OvcaFinder) and Zenodo (https://doi.org/10.5281/zenodo.10691378), which can only be used for non-commercial research purposes.

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

## Acknowledgements

This study was funded by the National Natural Science Foundation of China (Project No.62202403 for H.C., and 82171955 for X.L.), Natural Science Foundation of Guangdong Province (Project No.2021A1515012476 for X.L.) and Shenzhen Science and Technology Innovation Committee (Project No. SGDX20210823103201011 for H.C.).

## Author contributions

H.C. and X.L. conceived and designed the project; L.H.X., F.L., Y.C.L, T.T.D, J.C.Y., J.J.O., G.Q.L., X.R.H., S.L.H. and X.L. collected the data; J.Y.X., L.H.X., J.H.L. and H.C. analyzed the data; J.Y.X., J.H.L. and H.C. proposed the model; X.L., Y.C.L., X.L.L., T.T.D. and J.C.Y. conducted the reader study. J.Y.X., L.H.X. and F.L., wrote the paper. H.C., X.L., T.F.Z., X.W. and Y.L.L. revised the paper. All authors read and approved the final version of the article.

## Competing interests

The authors declare no competing interests.
