## [Peer Review File · Nature Communications]

Development and Validation of an Interpretable Model Integrating Multimodal Information for Improving Ovarian Cancer DiagnosisREVIEWERS' COMMENTS

Reviewer #1 (Remarks to the Author): expertise in ovarian cancer ultrasound diagnosis

The authors present a novel ML based prediction tool, which they call "Ovcafinder" and compare its performance to a DL tool based solely on image analysis and a "clinical" tool based solely on age, ca125 and lesion diameter. They conclude that the Ovca finder algorithm, which pulls in the three clinical parameters, the averaged ORADS score of 5 readers and the DL data, does better in predicting cancer compared to any of the other strategies alone.

The work is interesting in that there is significant interest in the use of ML or AI models in predicting disease, with active efforts underway for diagnostic imaging in particular.

I do not believe there has been a previously reported ML model that used a similar strategy, so in that regard, the report is novel. However, the validity of the conclusions and the actual clinical impact of the report is likely to be limited by several issues:

1. The authors correctly observe that many current ovarian risk stratification schemes such as ORADS ignore ca125 and other clinical information, despite their obvious role in clinical decision making, which has tended to make these schemes less clinically valuable. However, the authors in their approach effectively continue to ignore the clinical realities of how women with ovarian cancer typically present. Identifying ovarian cancer promptly in a woman with a CA125 of 10,000 or who presents with massive ascites is not difficult. Nobody needs a ML algorithm to help them with that. What would be of actual clinical value would be an algorithm that is highly predictive in the population of women with normal ca125s, relatively small masses (< 10 cm), who do not have other obvious clinical evidence of malignancy (eg, massive ascites, pleural effusion).
2. In general, the model should be referred to as a "predictive model" not a "diagnostic model" as diagnosis is not being established.
3. The fact that nearly half of the training set of 724 masses were malignant likely introduces significant bias limiting generalizability. In community based settings, typically < 5% of masses are malignant. Even in tertiary care referral centers in the US, the usual prevalence of malignancy is <25%. The authors should also clearly state in the text the % of masses that were malignant in the external dataset.
4. The inclusion of borderline tumors as malignant is problematic. While borderline tumors can recur it is exceedingly rare for a woman diagnosed with borderline tumor to die of disease. There is also no evidence to indicate that surgery by a gynecologic oncologist, or staging for that matter, has any impact on survival for women with borderline tumors. It should be clearly stated in the text what % of "malignant" masses were actually borderline. The authors need to show what the performance of the various strategies would be if only invasive ovarian/fallopian tube cancers are included, as that is the outcome where stage is linked to survival.
5. The exclusion criteria for which images were included lists "incomplete surgery or chemotherapy." The authors need to define what this is supposed to represent. A prior diagnosis of ovarian cancer should be an exclusion criteria, so it is unclear why any patients would have had "incomplete chemotherapy"
6. The vast majority of readers will not be experts in ML or AI algorithms. A more detailed description of terms used, including the RF models, what HITL means and how the human input was provided in the development of the model is needed.
7. In assessing ORADS, the readers showed a mean sensitivity of 96.2% and specificity of 73.3% in the internal dataset, and a mean sensitivity and specificity of 85.7% and 81.8%, respectively, in the external dataset. A difference between 96% and 86% sensitivity is significant. An explanation as to why performance varied so much between external and internal datasets is needed.
8. The unusual nature of the training set (with nearly 50% malignant/borderline masses) and the lack of detail about how human input was provided in the development makes it impossible to reproduce the work.

Reviewer #2 (Remarks to the Author): expert in machine learning pathology applications

The development of RF model serving as the OvcaFinder lacks details. Particularly unclear is the way of finding the N. How were the sets of different predictors identified? Please expand the description. Use Supplement if you need more room.

Receptive field of the DenseNet121 is 224x 224 pixels. Receptive fields of other DL models have similar size. How was the 512 x 512-pixel image fit into the respective model's receptive field?

Some heatmap visualizations in Figure 3B do not match original images. Moreover, order of images in columns is inconsistent. It seems that some columns contain Doppler images, some do not. In the caption, the authors wrote that "In the first row of each case ..." which suggests that cases are in columns but the images suggest that cases are in rows. Also, there is no low probability in the heatmap in D as the caption would suggest. Please unify/fix Figure 3 organization and clearly label each column and each row.

Figure 6. Image-based DL model performance level for sensitivity in row A is not marked.

In the introduction the authors mentioned that DL is criticized for acting as a black Box. However, they did not explain what DL that they trained specifically "finds" in ultrasound images and how that information can be used clinically. The explanation that "highlighted the most important areas that determined the model's decisions" is laconic and has been known to the machine learning community.

As the DL model significantly contributes to "decisions" made by the OvcaFinder decision, the authors should make additional efforts to explain the DL model behavior. For example, as localization and size of the hotspot feature gradient can vary from image to image, the authors may try explaining what the extent and localization of the hotspots mean in images of benign and malignant lesions. To provide more context, they can provide the percentage benign lesion images that display hotspots and the percentage of images without hotspots in images of cancerous lesions. Please explain if and how the extent of blood circulation could be a diagnostic factor that the model captures (compare for instance Figure 3A and Figure 3D).

RESPONSE TO REVIEWERS' COMMENTS

Dear Reviewers,

We thank you cordially for your time and efforts to review our manuscript. We have carefully studied your insightful comments and valuable suggestions, which help us significantly improve the quality of the revised manuscript. In addition, we have also made the contents of the paper clearer and easier to read. Specific changes are summarized as follows. Our response is in blue, and all changes to the revised manuscript are in red.

Yours Sincerely,

Authors of the manuscript

Reviewer #1 (Remarks to the Author): expertise in ovarian cancer ultrasound diagnosis

The authors present a novel ML based prediction tool, which they call “Ovcafinder” and compare its performance to a DL tool based solely on image analysis and a “clinical” tool based solely on age, ca125 and lesion diameter. They conclude that the OvcaFinder algorithm, which pulls in the three clinical parameters, the averaged ORADS score of 5 readers and the DL data, does better in predicting cancer compared to any of the other strategies alone.

The work is interesting in that there is significant interest in the use of ML or AI models in predicting disease, with active efforts underway for diagnostic imaging in particular.

I do not believe there has been a previously reported ML model that used a similar strategy, so in that regard, the report is novel. However, the validity of the conclusions and the actual clinical impact of the report is likely to be limited by several issues:

1. The authors correctly observe that many current ovarian risk stratification schemes such as ORADS ignore ca125 and other clinical information, despite their obvious role in clinical decision making, which has tended to make these schemes less clinically valuable. However, the authors in their approach effectively continue to ignore the clinical realities of how women with ovarian cancer typically present. Identifying ovarian cancer promptly in a woman with a CA125 of 10,000 or who presents with massive ascites is not difficult. Nobody needs a ML algorithm to help them with that. What would be of actual clinical value would be an algorithm that is highly predictive in the population of women with normal ca125s, relatively small masses (< 10 cm), who do not have other obvious clinical evidence of malignancy (eg, massive ascites, pleural effusion).

Response: Thanks for your comment. We included a total of 1,111 pathology-confirmed lesions in this retrospective, diagnostic study. Of those, 496 patients (496/1111, 44.6%) had normal CA125 levels, and 918 (918/1111, 82.6%) patients had lesions with a diameter less than 10cm. We found that the multimodal model OvcaFinder achieved the highest performance than the image-based DL model and clinical model, and significantly improved the performance of readers, which showed its certain clinical value to some extent. In future study, we will include more patients with normal CA125s, and relatively small lesions in a multicenter, and prospective setting.

2. In general, the model should be referred to as a “predictive model” not a “diagnostic model” as diagnosis is not being established.

Response: Thanks for your comments. To the best of our knowledge, a diagnostic model was used to identify whether an illness existed or not, but a predictive model aimed to forecast likely future outcomes. OvcaFinder was applied to discriminate benign from malignant adnexal lesions by using ultrasound images, human assessments, and readily-available clinical factors directly. To summarize, we thought it was more appropriate as “a diagnostic model” rather than “a predictive model”.

3. The fact that nearly half of the training set of 724 masses were malignant likely introduces significant bias limiting generalizability. In community based settings, typically < 5% of masses are malignant. Even in tertiary care referral centers in the US, the usual prevalence of malignancy is <25%. The authors should also clearly state in the text the % of masses that were malignant in the external dataset.

Response: Thanks for your comment. We had emphasized the relatively higher malignancy rate in the limitation. Both SYSUCC and CQCC were high-level cancer hospitals in China, where there were more patients with ovarian cancer. The followed-up cases were excluded for a lack of pathology results. Besides, we assigned borderline tumors into the malignant group in this study, which might result in a higher malignancy rate.

“First, there might be a selection bias in this retrospective study. Pathology-proven adnexal tumors from two cancer hospitals were enrolled, which resulted in a relatively higher malignancy rate than usual. A large-scale dataset, containing pathology-proven lesions, healthy controls and followed-up cases, not only from cancer hospitals but also general hospitals, would be useful for validating the DL model in a prospective setting to confirm its reliability.”

4. The inclusion of borderline tumors as malignant is problematic. While borderline tumors can recur it is exceedingly rare for a woman diagnosed with borderline tumor to die of disease. There is also no evidence to indicate that surgery by a gynecologic oncologist, or staging for that matter, has any impact on survival for women with borderline tumors. It should be clearly stated in the text what % of “malignant” masses were actually borderline. The authors need to show what the performance of the various strategies would be if only invasive ovarian/fallopian tube cancers are included, as that is the outcome where stage is linked to survival.

Response: Thanks for your advice. For the risk of recurrence, we assigned the borderline tumors in the malignancy group. We have added the percentages of borderline tumors in Table 1. In addition, in a series of high-quality researches, borderline tumors were regarded as malignant.

“Borderline tumours were assigned to the malignancy group.^{1,2,3}”

5. The exclusion criteria for which images were included lists “incomplete surgery or chemotherapy.” The authors need to define what this is supposed to represent. A prior diagnosis of ovarian cancer should be an exclusion criteria, so it is unclear why any patients would have had “incomplete chemotherapy”

Response: Thanks for your reminding. We have revised it as “(2) a prior diagnosis of ovarian cancer”.

6. The vast majority of readers will not be experts in ML or AI algorithms. A more detailed description of terms used, including the RF models, what HITL means and how the human input

was provided in the development of the model is needed.

Response: Thanks for your reminding. We have revised it as “OvcaFinder was a multimodal information-based model with human in the loop. Three clinical factors (patient’s age, lesion diameter, and CA125 concentration), O-RADS scores diagnosed by readers, and DL-based predictions were used to build the input with 5-dim vectors to develop OvcaFinder (Figure 1)”

7. In assessing ORADS, the readers showed a mean sensitivity of 96.2% and specificity of 73.3% in the internal dataset, and a mean sensitivity and specificity of 85.7% and 81.8%, respectively, in the external dataset. A difference between 96% and 86% sensitivity is significant. An explanation as to why performance varied so much between external and internal datasets is needed.

Response: Thanks for your advice. We have made more explanations about it. The enrollment of lesions with typical or non-typical features increased the diversity of our dataset. It is worth noting that the image-based DL models showed inferior performance in the external test set than that in the internal dataset, which confirmed the complexity of cases in the external set. But OvcaFinder performed well with an average AUC of 0.941 in the external dataset, which showed its generalization ability.

“This difference may be explained by distribution shift due to factors like relatively higher proportion of typical cases with heavier tumor burden in the internal test set, as evidenced by significantly higher CA125 levels ($p < 0.0001$)⁴.”

8. The unusual nature of the training set (with nearly 50% malignant/borderline masses) and the lack of detail about how human input was provided in the development makes it impossible to reproduce the work.

Response: Thanks for your reminding.

1. We must first admit that the high percentage of malignant cases in this diagnostic study was a limitation of our study. It was previously proved that screening was not effective in reducing mortality in ovarian cancer screening. However, accurate preoperative diagnosis was important for those who had adnexal lesions. If the lesions were malignant, they would go to the gynaecology oncology centre for further treatment to achieve better outcomes, while when the lesions were benign, exploratory surgery could be avoided to reduce unnecessary complications. We developed and evaluated the OvcaFinder by enrolling more malignant lesions in two cancer hospitals in China, which enabled the model to learn more features about cancerous disease in a diagnostic setting. We also found that OvcaFinder significantly improved the performance of readers. There is no doubt that prospective evaluation by enrolling more benign lesions and healthy controls is needed before its clinical application.

2. As for the detail about how human input was provided in the development, we have revised it as “OvcaFinder was a multimodal information-based model with human in the loop. Three clinical factors (patient’s age, lesion diameter, and CA125 concentration), O-RADS scores diagnosed by readers, and DL-based predictions were used to build the input with 5-dim vectors to develop OvcaFinder (Figure 1)” in the manuscript, and further provided the codes of the proposed model at the web repository of <https://github.com/Xiao-OMG/OvcaFinder>.

Reviewer #2 (Remarks to the Author): expert in machine learning pathology applications

The development of RF model serving as the OvcaFinder lacks details. Particularly unclear is the way of finding the N. How were the sets of different predictors identified? Please expand the description. Use Supplement if you need more room.

Response: Thanks for your reminding. We have revised it as “For each estimator, it used bootstrap method to randomly select a subset of the training dataset, which sample can be repeatedly selected. The subset was used to grow a decision tree. Therefore, we obtained a forest of N decision trees with different structures, as the trees were developed using different subsets. The majority voting algorithm was then used to combine the predictions of each decision tree to generate the final output. For the OvcaFinder and the clinical model, we both developed 291 RF models with different numbers of estimators ranging from 10 to 300. Finally, we found that N=70 for the OvcaFinder and N=20 for the clinical model would lead the models achieve the best performance of AUC on the validation dataset”

Receptive field of the DenseNet121 is 224x 224 pixels. Receptive fields of other DL models have similar size. How was the 512 x 512-pixel image fit into the respective model’s receptive field?

Response: Thanks for your reminding. All the models all have a pooling layer call “Global Average Pooling layer”, which can integrate the entire feature map information in to a 1 x 1 shape regardless of its size. Therefore, the 512 x 512-pixel image can be fit into the respective models.

Some heatmap visualizations in Figure 3B do not match original images. Moreover, order of images in columns is inconsistent. It seems that some columns contain Doppler images, some do not. In the caption, the authors wrote that “In the first row of each case ... ”which suggests that cases are in columns but the images suggest that cases are in rows. Also, there is no low probability in the heatmap in D as the caption would suggest. Please unify/fix Figure 3 organization and clearly label each column and each row.

Response: Thanks for your suggestions. We had revised it.

“Figure 3. Heatmap visualisation of image-based deep learning predictions of malignancy. Visual explanations of DL models are definitely important for qualitative review and clinical relevance, namely irregular solid components, projections, and areas with abundant blood flow signals. (A) Carcinosarcoma of a 44-year old female; (B) high-grade serous carcinoma of a 65-year old female; and (C) hydrosalpinx of a 49-year old female that was misdiagnosed by all readers but showed a low probability of malignancy in the heatmap. In the first row of each case, the first two images are B-mode images, and the following one is the colour Doppler image. The images in the the second row are their corresponding heatmaps.”

Figure 6. Image-based DL model performance level for sensitivity in row A is not marked.

Response: Thanks for your comment. In our study, to make a fair comparison, we compared the specificities via keeping similar sensitivities. That is the reason why the sensitivity was not significantly different. And we have stated it in the manuscript.

In the introduction the authors mentioned that DL is criticized for acting as a black Box. However, they did not explain what DL that they trained specifically “finds” in ultrasound images and how that information can be used clinically. The explanation that “highlighted the most important areas that determined the model’s decisions” is laconic and has been known to the machine learning community.

As the DL model significantly contributes to “decisions” made by the OvcaFinder decision, the authors should make additional efforts to explain the DL model behavior. For example, as localization and size of the hotspot feature gradient can vary from image to image, the authors may try explaining what the extent and localization of the hotspots mean in images of benign and malignant lesions. To provide more context, they can provide the percentage benign lesion images that display hotspots and the percentage of images without hotspots in images of cancerous lesions. Please explain if and how the extent of blood circulation could be a diagnostic factor that the model captures (compare for instance Figure 3A and Figure 3D).

Response: Thanks for your suggestions. We have revised it. As for blood circulation, after reviewing these images and hotspots, we found that blood signals in irregular solid component can be a diagnostic factor. For example, in Figure 3C, the blood signal can be found in the regular normal ovary tissues. In Figure 3B, the blood signal alongside the lesion was also not highlighted.

“As shown in Figure 3, the red regions of the heatmaps contributed most to a given classification, while the blue regions were less important. To be more specific, areas with irregular solid components or projections on B-mode images were highlighted in the heatmap and were valuable features for malignancy prediction. With regard to colour Doppler images, the heatmap focused on areas with abundant angiogenesis. These were consistent with the diagnostic criteria of ovarian tumors in clinical practice. For benign lesions, there were 27.8% (15/54) and 19.8% (60/306) cases with hotspots shown in the internal and external test sets, respectively. As for cancerous lesions, a percentage of 4.0% (3/75) in the internal test set and 12.3% (10/81) in the external test set were observed without hotspots displayed, respectively.”

1. Guo Y, *et al.* Deep learning with weak annotation from diagnosis reports for detection of multiple head disorders: a prospective, multicentre study. *The Lancet Digital health* **4**, e584-e593 (2022).
2. Van Calster B, *et al.* Validation of models to diagnose ovarian cancer in patients managed surgically or conservatively: multicentre cohort study. *bmj* **370**, (2020).
3. Chen H, *et al.* Deep Learning Prediction of Ovarian Malignancy at US Compared with O-RADS and Expert Assessment. *Radiology* **304**, 106-113 (2022).
4. Ayhan A, *et al.* Metastatic lymph node number in epithelial ovarian carcinoma: does it have any clinical significance? *Gynecologic oncology* **108**, 428-432 (2008).

REVIEWERS' COMMENTS

Reviewer #1 (Remarks to the Author):

The authors have done a reasonable job of addressing concerns, in particular acknowledging the bias resulting from having the training set derived from a very high risk population. I would like them to state this limitation more plainly, however, such as "The applicability of the strategy to lower risk populations where the prevalence of cancer is low remains to be determined."

It is also important to know the characteristics of the cancers "detected" by the system for readers to judge the potential clinical utility. Therefore, there needs to be 1-2 sentences that specifically describe the characteristics of the CANCERS (not the median characteristics of the entire set, which is done in Table 1), specifically, the median size and range of the cancers and the % presenting with elevated ca125, ascites or evidence of metastatic disease on imaging (carcinomatosis, pleural effusion, liver mets, adenopathy), stage, histology and % borderline. If a high percentage of the cancers "detected" by the system were clinically alarming due to the presence of these characteristics then the expected utility of the strategy is less. It is currently not possible to know this information from Table 1.

Reviewer #2 (Remarks to the Author):

Thank you for addressing my comments. The methodology and results are sound. A few new comments are below:

- 1) Please cite original paper(s) describing the SHAP technique that helps explain feature ranking and importance by the RF model.
- 2) Ln. 315, you wrote "did not clarify the most predictive features identified". As the other reviewer noted, please be careful about the context of using "predictive" word. In this sentence, or elsewhere in the manuscript, the authors should explain what "predictive feature" means or refers to in the context of this study.
- 3) Ln. 170, there are two comas, and the sentence is unclear.

RESPONSE TO REVIEWERS' COMMENTS

Reviewer #1 (Remarks to the Author):

The authors have done a reasonable job of addressing concerns, in particular acknowledging the bias resulting from having the training set derived from a very high risk population. I would like them to state this limitation more plainly, however, such as "The applicability of the strategy to lower risk populations where the prevalence of cancer is low remains to be determined."

Response: Thanks for your advice. We have added it in the limitation.

We acknowledge the limitations of this study. First, there might be a selection bias in this retrospective study. Pathology-proven adnexal tumors from two cancer hospitals were enrolled, which resulted in a relatively higher malignancy rate than usual. The applicability of the strategy to lower risk populations where the prevalence of cancer is low remains to be determined. A large-scale dataset, containing pathology-proven lesions, healthy controls and followed-up cases, not only from cancer hospitals but also general hospitals, would be useful for validating the DL model in a prospective setting to confirm its reliability.

It is also important to know the characteristics of the cancers "detected" by the system for readers to judge the potential clinical utility. Therefore, there needs to be 1-2 sentences that specifically describe the characteristics of the CANCERS (not the median characteristics of the entire set, which is done in Table 1), specifically, the median size and range of the cancers and the % presenting with elevated ca125, ascites or evidence of metastatic disease on imaging (carcinomatosis, pleural effusion, liver mets, adenopathy), stage, histology and % borderline.

If a high percentage of the cancers "detected" by the system were clinically alarming due to the presence of these characteristics then the expected utility of the strategy is less. It is currently not possible to know this information from Table 1.

Response: Thanks for your advice. We have added the characteristics of malignant tumors (% of borderline tumors, median size and range, the percentage of ascites and peritoneal thickening or nodules) in the Results part. We must admit that cancers collected from two cancer hospitals were relatively advanced. In clinical practice, nearly 75% patients were diagnosed at late stages at initial diagnosis, and this is the reason why we need to make more effort to achieve early and accurate diagnosis of ovarian cancer. As you suggested, we have added "The applicability of the strategy to lower risk populations where the prevalence of cancer is low remains to be determined" in the limitation, and the diagnostic value of OvcaFinder in average-risk women and early-stage ovarian cancer must be further investigated.

"Among 509 malignant lesions, there were 57 borderline tumors (11.2%). For malignant lesions, the average lesion diameter was 83.4mm (range: 13 to 225mm). Taking 35 U/mL as threshold, nearly 88.2% (449/509) patients had evaluated CA125 levels. Ascites and peritoneal thickening or nodules were found in 272 and 306 patients in ultrasound images, respectively."

Reviewer #2 (Remarks to the Author):

Thank you for addressing my comments. The methodology and results are sound. A few new comments are below:

1) Please cite original paper(s) describing the SHAP technique that helps explain feature ranking and importance by the RF model.

Response: Thanks for your advice. We have already cited it in the manuscript.

With OvcaFinder, readers significantly improved their diagnostic performance and decreased their false positives. In addition to identifying ovarian cancer, OvcaFinder is able to give explanations to its predictions by highlighting the most important areas in heatmaps and reveal the impact of each parameter with Shapley values¹.

2) Ln. 315, you wrote “did not clarify the most predictive features identified”. As the other reviewer noted, please be careful about the context of using “predictive” word. In this sentence, or elsewhere in the manuscript, the authors should explain what “predictive feature” means or refers to in the context of this study.

Response: Thanks for your comment. First, we have removed the word “predictive”, and revised this sentence into “Most DL models built previously for adnexal tumour diagnosis from ultrasound images did not show the most important features or areas that were highly relevant to their final classification, which hinders the building of trust that readers have in DL models. ”.

Second, we have explained the most informative features of the heatmap in the Results (As shown in Figure 1, the red regions of the heatmaps contributed most to a given classification, while the blue regions were less important. To be more specific, areas with irregular solid components or projections on B-mode images were highlighted in the heatmap and were valuable features for malignancy prediction. With regard to colour Doppler images, the heatmap focused on areas with abundant angiogenesis.), and the Discussion part (Here, we found that heatmaps facilitated the assessment of adnexal masses by highlighting areas with irregular solid components, projections, or abundant blood signals, which is in accordance with current guidelines.).

Third, we also provided a series of cases along with their corresponding heatmaps in Figure 1 to explain what the DL model has learned from ultrasound images.

3) Ln. 170, there are two comas, and the sentence is unclear.

Response: Thanks for your comment. We have revised it.

For each estimator, we use Bootstrapping method to randomly resample the training set with replacement 1,000 times to create simulated datasets. A simulated dataset was used to grow a decision tree. Therefore, we obtained a forest of N decision trees with different structures, as the trees were developed using different simulated datasets.

1. Lundberg SM, Lee S-I. A unified approach to interpreting model predictions. In: *Proceedings of the 31st International Conference on Neural Information Processing Systems*. Curran Associates Inc. (2017).

The value of multimodal information has not been fully investigated in ovarian cancer diagnosis. Here, we present OvcaFinder, an interpretable model constructed from ultrasound images-based deep learning (DL) predictions, Ovarian–Adnexal Reporting and Data System scores from radiologists, and routine clinical variables. OvcaFinder outperforms the clinical model and the DL model with area under the curves (AUCs) of 0.978, and 0.947 in the internal and external test sets, respectively. With the help of the OvcaFinder, radiologists improved inter-reader agreement, decreased their false positives, and increased their diagnostic performances with AUCs from 0.927 to 0.977 and from 0.904 to 0.941, and in the internal and external test sets, while maintaining similar sensitivities, respectively. This highlights the potential of OvcaFinder to serve as a non-invasive tool to improve the accuracy, and consistency of radiologists in ovarian cancer diagnosis.